# Magnesium Ortho-Vanadate/Magnesium Oxide/Graphene Oxide Embedded through Cellulose Acetate-Based Films for Wound Healing Applications

**DOI:** 10.3390/ma16083009

**Published:** 2023-04-11

**Authors:** Fatemah A. Taher, Mohamed Gouda, Mai M. Khalaf, Saad Shaaban, Alnoor Y. A. Al Bosager, Dania A. A. Algafly, Metwally K. Mahfouz, Manal F. Abou Taleb, Hany M. Abd El-Lateef

**Affiliations:** 1Department of Chemistry, College of Science, King Faisal University, Al-Ahsa 31982, Saudi Arabia; 218004076@kfu.edu.sa (F.A.T.); mmkali@kfu.edu.sa (M.M.K.); sibrahim@kfu.edu.sa (S.S.); 2Chemistry Department, Faculty of Science, Sohag University, Sohag 82524, Egypt; 3Department of Chemistry, Faculty of Science, Mansoura University, Mansoura 35516, Egypt; 4Alkifah Academy Private School, Al-Ahsa 31982, Saudi Arabia; saqeryanoor@gmail.com (A.Y.A.A.B.); daniaalghafli@gmail.com (D.A.A.A.); 5Department of Biochemistry, Animal Health Research Institute, Sohag Branch, Agriculture Research Center, Sohag 82524, Egypt; 6Department of Chemistry, College of Science and Humanities, Prince Sattam Bin Abdulaziz University, Al-Kharj 11942, Saudi Arabia

**Keywords:** magnesium ortho-vanadate, magnesium oxide, graphene oxide, cellulose acetate, wound healing

## Abstract

A multifunctional nano-films of cellulose acetate (CA)/magnesium ortho-vanadate (MOV)/magnesium oxide/graphene oxide wound coverage was fabricated. Through fabrication, different weights of the previously mentioned ingredients were selected to receive a certain morphological appearance. The composition was confirmed by XRD, FTIR, and EDX techniques. SEM micrograph of Mg_3_(VO_4_)_2_/MgO/GO@CA film depicted that there was a porous surface with flattened rounded MgO grains with an average size of 0.31 µm was observed. Regarding wettability, the binary composition of Mg_3_(VO_4_)_2_@CA occupied the lowest contact angle of 30.15 ± 0.8^o^, while pure CA represents the highest one at 47.35 ± 0.4°. The cell viability % amongst the usage of 4.9 µg/mL of Mg_3_(VO_4_)_2_/MgO/GO@CA is 95.77 ± 3.2%, while 2.4 µg/mL showed 101.54 ± 2.9%. The higher concentration of 5000 µg/mL exhibited a viability of 19.23%. According to optical results, the refractive index jumped from 1.73 for CA to 1.81 for Mg_3_(VO_4_)_2_/MgO/GO@CA film. The thermogravimetric analysis showed three main stages of degradation. The initial temperature started from room temperature to 289 °C with a weight loss of 13%. On the other hand, the second stage started from the final temperature of the first stage and end at 375 °C with a weight loss of 52%. Finally, the last stage was from 375 to 472 °C with 19% weight loss. The obtained results, such as high hydrophilic behavior, high cell viability, surface roughness, and porosity due to the addition of nanoparticles to the CA membrane, all played a significant role in enhancing the biocompatibility and biological activity of the CA membrane. The enhancements in the CA membrane suggest that it can be utilized in drug delivery and wound healing applications.

## 1. Introduction

The body’s barriers in defiance of pathogens start with the skin organ that preserves the homeostasis of living organisms’ bodies [1,2,3,4]. Any severe damage to skin layers may have an influence on human health [5]. Tissue engineering’s purpose is to develop the damaged skin regeneration process [5]. One of the main operational treatments for skin wound regeneration is to enhance angiogenesis through the healing progression [6,7,8,9]. Likewise, discovering a bio-scaffold that performs as a natural extracellular matrix has proven important [10]. Over the latest decade, there has been remarkable progress in nano-biomaterials usage in the tissue engineering medical sector. Countless approaches have been introduced to boost the rate and superiority of the healing progression with angiogenic nano-materials [5]. Undeniably, skin-wound healing is a complicated process [11,12]. It happens through many stages, i.e., the stopping of blood flow (hemostasis), soring, inflammation, proliferation, and restoration, according to the intensity of the damage [13,14]. Anti-inflammatory performance and surface design are considered important healing factors [6,7,8,9].

Moreover, the delineation of the grafting material must offer an interconnected surface to imitate extracellular structure [6,15]. Additionally, polymers were assessed as a raw matrix for bio-scaffold materials owing to their biocompatibility. In addition, several bio-polymeric materials have induced tissue integration [16,17,18]. Among these, the cellulosic material is considered accessible because it originated naturally. The three active hydroxyl groups/one molecule are dynamic in introducing cellulosic proper physicochemical features [19,20,21]. Additionally, cellulose esterification produces cellulose acetate, the most prevalent natural polysaccharide in the environment, a biodegradable thermoplastic polymer [22,23,24]. CA is a semi-synthetic, biocompatible polymer with convenient hydrophilicity, degradability, and moisture retention [25]. Additionally, it improves the biological contact between fibroblasts and scaffolds and is very soluble in organic solvents [25]. Due to its biocompatibility, chemical stability, low toxicity, and moderate mechanical strength, CA has been used in a wide range of applications, including paper, membranes, water filtration, and biological applications [26].

Indeed, NPs inclusion in polymeric matrix offers better bio-applicability [19]. More in-depth research on the mechanisms of metals in the healing process aims at permeability and surface morphology adjustment, manipulation of crystals’ active sites, and surface roughness, porosity, and elasticity, so as to achieve a fast and safe healing process. The significant metal-based scaffolds’ biological activity comprises the germicidal action during the wound treatment process. Magnesium is a vital mineral that is the 4th most common trace element in the mammalian body. Similarly, it boosts the catalytic activity of about 100 enzymes. As well as this, Mg is crucial in improving the mechanical behavior of bone, besides its important role in skeletal metabolism [27,28]. Additionally, since 1995, vanadium hass proven effective in treating diabetes [29,30]. In addition, sorption shows significant enhancement in vanadium composites [29,30]. Unfortunately, V-concentrations that exceed 0.01 × 10^−3^ M cause the accumulation of vanadium in tissues [31]. Vanadium exists in a digestive canal in the form of V^4+^ up until reached the duodenum, nonetheless, V^5+^ is absorbed smoothly in the intestine (with four times the absorption capability of V^4+^) [32,33]. Sequentially, vanadium redox response is crucial in its applicability. And vanadate composition is able to interact with cellular proteins as phosphatases [34].

Finally, vanadium occurrence in the human organs, by ng/g wet weight, is: liver 7.5; thyroid gland 3.1; kidney 3; lung 2.1; heart 1.1; finally, fat and muscle 0.55. Additionally, vanadate causes diverse effects as a function of contact level-dependent, small concentrations of vanadate, which retain a mitogenic behavior, to a maximum amount at 100 µM, acting as a growth aspect [35]. However, in high concentrations vanadate displays clear declines in cell viability and proliferation proposing that cytotoxicity is owing to reactive oxygen species released [35]. Structurally, graphene oxide is composed of a 2D layer whose thickness is equivalent to a carbon atom’s radius [36,37,38,39]. GO displays extraordinary mechanical, electrical, and biocompatible behaviors [40,41,42]. The highly oxygenated function groups offer highly active surfaces [43,44,45]. These oxyanions release induce an antibacterial performance [46], besides recommending its usage in loading and releasing drugs [47,48,49]. GO displays remarkable biological properties, such as biocompatibility and antibacterial capabilities, as well as remarkable mechanical and electrical properties. The GO may also have a moderate antibacterial effect, according to both physical and chemical trends. In other words, the release of ROS through the microbial environment is attributed to the antibacterial action of GO and is categorized as a chemical strategy, whereas direct contact with the edges of the GO nanosheets introduces a physical strategy to degenerate bacterial cells. Additionally, the surface hydroxyl group containing oxyanions provides a large number of active sites and may be thought of as a source of ROS release. These oxygen-containing groups also encourage a high capacity for drug loading and release.

Therefore, merging these components upon their functions may introduce wound healing bio-scaffold. The physical characteristics, such as morphological, structural, thermal, and optical performances of the acquired nano-films, will be inspected, besides the cellular response towards the cell line of the human lung in vitro. 

Polymeric films play a significant role in the healing process, as it is the matrix in which the nanoparticles are encapsulated. When the fabricated film is placed over the wounded area, the film releases the nanoparticles. These nanoparticles play a significant role in cell proliferation and cell growth. On the other hand, bacterial invasion plays a key role in delaying the healing time so that the role which encapsulating nanoparticles play in antibacterial activity is important to accelerate the healing process. Furthermore, the addition of nanoparticles and the porosity of the films play a vital role in decreasing the contact angle, which means that the hydrophilic nature of the film is increased. This leads to an increase in the chemical activity of the film and increases in cell attachment which accelerates the healing process. Although this scaffold has unique properties, the chemical stability of the scaffold could not be controlled. The rate of degradation of the scaffold and drug release rate is not adjustable. It is very important to adjust the ionic release dose as the high dose may cause cytotoxicity while the low released dose may have a weak effect on the healing process. Therefore, it is very important to adjust the release the rate of ionic release.

The aim of this work is to design membranes from CA to be utilized in wound healing and drug delivery applications. Several nanoparticles have been added to the CA to enhance its properties and also to give it several more properties. Mg is chosen to be the additive in the membrane because it has appropriate mechanical properties and because it enhances the catalytic activity of many enzymes. On the other hand, the presence of vanadate in the scaffold is due to its ability to enhance the interaction with cellular proteins and its antibacterial activity. Meanwhile, GO exhibits suitable properties for wound healing applications such as its antibacterial activity, drug-loading capacity, mechanical properties, electrical properties, and hydrophilic behavior. The unique properties of these nanoparticles make it a good choice to be embedded in the CA membrane for wound healing applications.

## 2. Experimental Techniques

### 2.1. Preparation of Scaffold with Different Contents of Oxides

Magnesium ortho-vanadate/ magnesium oxide and CA are bought from PubChem. Additionally, GO is prepared in the laboratory by a modified Hummers’ method [36,39]. Moreover, the CA films were formed with the casting methodology. To prepare the films, about 10 g of CA powder were dissolved in 100 mL of acetone under stirring as stock solution. The first sample was pure CA film, so about 20 mL of CA solution was poured into a Petri dish to dry the polymeric solution at a drier furnace. On the other hand, the second sample was prepared by injecting 20 mL of CA in a bottle under magnetic stirring then about 0.25 g of Mg_3_(VO_4_)_2_ was wisely added to the solution. After an hour of stirring, the Mg_3_(VO_4_)_2_ nanoparticles have been well dispersed in the solution. Then the second sample was poured into a Petri dish and dried. Furthermore, the third sample was obtained by adding 0.25 g of MgO nanoparticles to 20 mL of CA solution and stirring for 60 min. After that, the solution was poured into a Petri dish and dried. Moreover, the fourth sample was prepared by adding 0.125 g of Mg_3_(VO_4_)_2_ to the polymeric solution under stirring, followed by approximately 0.125 g of MgO added to the same solution. After dispersing the nanoparticles in the CA solution, the solution was poured into a dish and dried. Finally, the last sample was prepared with three additives to 20 mL of CA. These additives 0.1 of Mg_3_(VO_4_)_2_, 0.1 g of MgO, and 0.05 g of GO were separately added to the CA solution separately as dispersed using magnetic stirring. After that, the solution was poured and dried to obtain the film. The name of the cellulose acetate film was FAC. 

### 2.2. Material Characterization

The X-ray diffractometer model was used to perform the X-ray diffraction (XRD) investigations (Cu Kα radiation, 45 kV, 40 mA, Malvern, WR14 1XZ, Worcestershire, UK, analytical-x’ pertpro). The primary purpose of it was to examine the phase composition of the fabricated films’ composition. All XRD curves were scanned in 5° ≤ 2θ ≤ 70° scope with a step size of 0.02° and a step time of 0.5 s. The surface morphology was investigated using scanning electron microscopy (SEM, ZEISS model, Anhui, China) at a high voltage of 8 kV. The cross-section images were taken using FESEM (model: QUANTA-FEG250, Kolkata, WB, India). It could be noticed that each sample was scanned using two different magnifications and different positions. TGA was performed using a thermal analyzer (DTG-60H SHIMADZU, Kyoto, Japan) with an airflow rate of 100 mL/min from ambient temperature up to 600 °C. There was a 10 °C/min heating rate. The optical characteristics of polymeric samples were examined using UV-visible (UV-Vis, Spectrophotometer model UV-9000S, Shanghai Metash Instruments Co., Shanghai, China). The wettability of the films has been measured using a custom system. Approximately 2 cm × 2 cm from the sample was placed in a holder in front of the camera. After this, a water droplet was dropped over the sample the image was taken by (HiView) application. Using this program, the contact angle was measured by drawing a baseline measuring the angle from left and right. The average and standard deviation was calculated using excel functions. The Raman was detected using a laser beam (532 nm) on the film’s surface.

The ionic concentration was detected using an ionic chromatography system. Firstly, 0.1 g of the composition of (VO_4_)_2_/MgO/GO@CA was immersed in double distilled water for 24 h. Then, 10 µL was taken from the solution to be diluted with 2 mL of double distilled water. Finally, 1 mL from the previous solution was injected through the system to be detected. The experiment was done for each element individually.

### 2.3. In Vitro Cell Viability Tests

To examine cell viability, the normal lung cells were cultured in Dulbecco’s modified Eagle’s medium (DMEM, Gibco).

To examine cell viability, the normal lung cells (A138) were cultured in Dulbecco’s modified Eagle’s medium (DMEM, Gibco). The examination of cell viability was undertaken by VACSERA Co. in Giza, Egypt. Furthermore, the normal lung cell lines were isolated from the lung tissue of a 3-month-old female embryo. Accession Numbers: WI-38 (ATCC CCL-75). On the films through 24-well plates, cells were cultivated at a density of 5 × 10^3^ (cells/cm^2^), and they were then incubated at 37 °C. After three days of incubation, the media was removed, MTT (3-(4,5-dimethylthiazol-2-yl)-2,5-diphenyltetrazolium bromide) was injected into each well, and the vitality of the cells was assessed using an optical analyzer. On the films through 24-well plates, cells were cultivated at a density of 5 × 10^3^ (cells/cm^2^), and they were then incubated at 37 °C. After three days of incubation, the media was removed, MTT (3-(4,5-dimethylthiazol-2-yl)-2,5-diphenyltetrazolium bromide) was injected into each well, and the vitality of the cells was assessed using an optical analyzer. The data of cell viability was drawn using the origin lab and the viable cells ratios were taken from the graph. The study of cell viability was repeated three times, and the standard deviation was calculated and plotted.

## 3. Results and Discussions 

### 3.1. Qualitative and Quantitative Structural Investigation

The XRD data represents cellulose acetate-based films. The main broad and intense crest at 2θ = 20.5° is stated as the CA composition existence [13]. The characteristic peaks of cubic MgO in binary film MgO@CA diffractogram are centered at 2θ = 43.1° and 62.5°, which are associated with (200) and (220) planes, respectively [50]. The crests centered at 32.6, and 35.7° are characteristic of Mg_3_(VO_4_)_2_ composition [51]. The merging of GO upon ternary film is confirmed by a slight deviation in the revealed peaks, as well as the fact that the GO characteristic peak at 10.6° does not appear at 5. This may be explained by the disturbance of the 2D structure of GO upon film formation. Concisely, the XRD diffractogram supports the studied ingredient’s involvement (Figure 1).

Regarding FTIR spectra, the CA presence is confirmed by bands at 1024, 1219, 1374, and 1741 cm^−1^ which are equivalent to C–O–C, C–O–C, C–methyl group, and C–O, respectively. Referring to magnesium vanadate, the strong band centered at 1750 cm^−1^ and the weak band at 1654 cm^−1^ overtones of VO terminal [52]. Other than that, MgO characteristic peak at 890 cm^−1^ is recognized as Mg–O vibration [53]. The variations in peak intensities confirm the different insertions to CA film and film compatibility. The peak at 1510 cm^−1^ reflects absorbed water. Moreover, the peaks ascribe GO include the peaks positioned at wave number 3480 cm^−1^ are equivalent to the adsorbed H_2_O on GO, besides the bands assigned to Ʋ C-C at 1658 cm^−1^ [54]. Thus, Figure 2 represents the FTIR spectra of the five films that confirm the chemical structure of each.

The quantitative and qualitative elemental results of the Mg_3_(VO_4_)_2_/MgO/GO@CA film are collected via energy dispersive X-ray (EDX) analysis. The quantitative investigations are detected in the function of signal intensities [55]. The tabulated data confirms the detection of C, N, O, Mg, and V with 50.66, 2.59, 44.77, 1.81, and 0.16%, as in Table 1. Unquestionably, oxygen occupied the second highest atomic % after carbon because of it being bound to all elements separately at 0.5 keV, as shown in Figure 3 [56]. 

### 3.2. Morphological Study

The CA-based binary, ternary, and quaternary nanofilms morphological features are displayed in Figure 4. The binary composition of Mg_3_(VO_4_)_2_@CA film displays magnesium orthovanadate as anthropomorphic rods which spread upon CA matrix. These grains appear separately and in small aggregates, with an average width of 0.11 µm, and an average length of 0.55 µm, as shown in Figure 4a. Moreover, Mg_3_(VO_4_)_2_/MgO@CA film appears as a cracked flattened CA media with scattered oxy-metallic ingredients upon the surface. MgO displays oval scattered grains (0.18 µm), while magnesium orthovanadate appears as a medium aggregate with an average size of 0.85 µm. The Mg_3_(VO_4_)_2_/MgO@CA surface shows as smoother than the two other screened compositions. Furthermore, Mg_3_(VO_4_)_2_/MgO/GO@CA film introduces a non-deep porous structure with flattened rounded grains upon adding graphene oxide to the ternary composition. The well-distributed MgO grains upon the polymeric matrix show an average size of 0.31 µm. The demonstrated micrographs exhibit advances in structural integrity and porosity that offer better biocompatibility. In Figure 4(d1–e2), it is obvious that the nanoparticles are immersed and distributed inside the film. The agglomeration of the nanoparticles in Figure 4(d1,d2), is decreased as shown in Figure 4(e1,e2) may be because of the decrease in the concentration of both MgO and Mg_3_(VO_4_)_2_ from 0.125 g for each to 0.1 g for each. Cell viability is boosted by many factors; the film’s porosity which is related to the improvement of cell feeding and adhesion degree of attachment, the uniform spread of dopants within the matrix, particle size, as well as surface roughness [57]. YU et al., 2018 reported that sodium orthovanadate (SOV) inhibited the viability of the 8505C cell lines (human anaplastic thyroid carcinoma cells) with mean IC50 values of SOV 2.30 µM [58]. Thus, from the previous survey and the introduced topological feature, the quaternary nano-film is promising in biological usage.

### 3.3. Wettability Properties

Irrefutably, the more contracted contact angle, the higher the adhesion property, and also the better bio-applicability. The contact angle was measured for all samples. The pure CA film exhibited a contact angle of 47.35 ± 0.45°, while the angle decreased to 30.15 ± 0.75° after the addition of Mg_3_(VO_4_)_2_ to the CA film. On the other hand, the contact angle decreased to 35.55 ± 0.25° after the addition of MgO to the pure CA. Moreover, the contact angle of the CA film was decreased to 37.84 ± 0.30° when Mg_3_(VO_4_)_2_ and MgO were added to the pure CA. Further, the combination of Mg_3_(VO_4_)_2_, MgO, and GO decreased the contact angle of CA to 35.8 ± 0.4°. The binary composition of Mg_3_(VO_4_)_2_@CA occupies the lowest contact angle, while pure CA represents the highest one, as shown in Figure 5. Thus, all dopants lead to the formation of higher wettability nano-films. The insertion of magnesium oxy-compositions and GO nonmetric made the fabricated scaffold more hydrophilic. The wettability behavior of these films recommends their usage of it biologically. It can be noticed that the addition of GO to Mg_3_(VO_4_)_2_/MgO@CA has decreased the contact angle and increased the hydrophilic behavior of the film. This could be due to the hydrophilic nature of GO. According to the surface morphology results, it can be noticed that the porosity has increased in the last sample after the addition of GO to the film, and also that the surface roughness was high in the case of the addition of Mg_3_(VO_4_)_2_ to the CA film. The roughness and porosity play a key role in increasing the hydrophilicity of the polymeric films. This may explain the low contact angle of the second and final samples. S.I. Al-Saeedi et al. encapsulated copper oxide into CA scaffolds for wound healing applications [26]. It is reported that the addition of copper oxide to the scaffold played a significant role in changing the wettability properties of CA [26]. It becomes more hydrophilic after the addition of copper oxide, with an angle of 85.4° [26].

### 3.4. Cell Viability Percentage upon the Usage of Mg_3_(VO_4_)_2_/MgO/GO@CA against Normal Lung Cells

Herein, the evaluation of Mg_3_(VO_4_)_2_/MgO/GO@CA film cytotoxicity against normal lung cells. The observed cell viability % amongst the usage of 4.9 µg/mL of Mg_3_(VO_4_)_2_/MgO/GO@CA is 95.77 ± 3.2%, while 2.4 µg/mL hits 101.54 ± 2.9%. The higher concentration of 5000 µg/mL shows a viability of 19.23 ± 3.1%, as shown in Figure 6. Iordachescu et al., 2002, have observed vanadate with diverse effects as a function of exposure level. Low concentrations of vanadate possess a mitogenic behavior, to a maximum extent at 100 µM, acting as a growth factor. However, in high concentrations vanadate shows clear declines in cell viability and proliferation proposing that cytotoxicity is owing to reactive oxygen species released [35]. Through wound healing, epidermal growth factor (EGF), platelet-derived growth factor (PDGF), and fibroblast growth factor (FGF) serve a vital role [59]. Briefly, vanadate stimulates cell proliferation and bone collagen production in vitro. Besides this, vanadate has been shown to mimic the effects of fibroblast growth factor in that it inspires endothelial cells to occupy collagen matrices and organize into tubules [59]. Indeed, the interaction mechanism between the wounded tissues and the implant film is firmly linked to the topological features of the tested composition, measured as porosity [60]. The high porous surface enhances the circulation of metabolites and vascularization. Likewise, roughness degree and wettability directly affect the film’s biological performance. In addition to this, the Mg ions’ role is to boost metabolic reactions. Consequently, controlling the composition and design of films may introduce a biomaterial with desirable behaviors. S. I. Al-Saeedi et al. encapsulated copper oxide into a CA scaffold for wound healing applications [26]. The results showed that by changing the concentration of copper oxide, the ratio of viable cells also changes [26]. The lowest cell viability ratio was obtained from the sample of the lowest concentration of copper oxide with a cell viability rate of 91.3 ± 4%, while the highest cell viability ratio was obtained from the sample of the highest concentration of copper oxide with a cell viability rate of 96.4 ± 4% [26]. L. Lei et al. loaded trilazad mesylate into a CA scaffold with different concentrations for wound healing applications [61]. The highest cell viability ratio was obtained from the sample cellulose acetate/trilazad mesylate 3%. After 7 days of culturing this sample, the cell viability reached 75.19 ± 3.28%. On the other hand, the sample reached 94.79 ± 2.61% after 14 days of culturing [61]. J. Prakash et al. prepared CA scaffolds doped with GO, TiO_2_, or curcumin for wound healing application [13]. It is reported that the pure CA scaffold has the highest cell viability ratio near 97% [13]. On the other hand, the addition of GO to the CA scaffold decreased the cell viability ratio to almost 91%. Further, the addition of GO and TiO_2_ to the scaffold gives a cell viability ratio of nearly 88% [13]. Moreover, the cell viability reached almost 91.5% after the addition of curcumin to the previous sample [13]. The concentration of the nanoparticles compared to the polymeric film is very low as the total concentration of the nanoparticles is 0.25 g. the ratio between the nanoparticles and the film is 0.25/2 w/w. This low concentration of nanoparticles is not toxic to the human body. To make sure that the sample is not toxic to the human body, the films are cultured to test the cell viability. According to the previous results of cell viability the cells not only safe and alive but also are growing and proliferating. This means that the concentration of nanoparticles is safe for the human body.

### 3.5. Thermal Study

Thermogravimetric analysis (TGA) is done for revealing the thermal behavior of Mg3(VO4)2/MgO/GO@CA. The highly oxygenated film is exposed to temperature (30–600 °C) at a rate of 10 °C/min. The thermal degradation comprises three stages, (30:289 °C/temperature peak (Ts) 64.5 °C/Wt. loss 13%), (289:375 °C/Ts 353.2 °C/Wt. loss 52%), and (375:472 °C/Ts 463 °C/wt. loss 19%), leaving a residue of about 16%. Mainly, the mass loss of 13% occurred up to 289 °C, assigned to H_2_O attached to CA hydroxyl groups. After that, there were two obvious stages of degradation as previously stated. The first step involves mass loss of 52% which is assigned to CA chain disintegration owing to the disruption of glycosidic bonds [22]. The second involves a mass loss of 19%, which reflects CA carbonization, resulting in the complete degradation of CA [22]. According to Figure 7, the two main degradation steps are included in the MgO ingredient reduction step and the oxidation step. The wt. loss % of the reduction step is tightly related to oxygen pressure and the higher O_2_ Pressure causes a decline in wt. loss [62]. The MgO crystallizing water loss occurs in the scope from 50 to 370 °C [53]. Moreover, at high temperatures, higher oxygen vacancies are found [63]. On the other hand, Mattevi et al. proposed that the influence of heating on GO at 450 °C is equivalent to the GO reduction process by hydrazine monohydrate at 80 °C shadowed by sintering at 200 °C [64]. As well as this, the hydrogen bonding between film ingredients plays a stabilizing role [65]. Furthermore, the compact structure (films) of CA-based quaternary film offers a larger number of polymeric chains, and, also, an electrostatic interactions occurrence [65]. The previous explanation discusses the relatively stable nano-film.

### 3.6. Optical Behavior

Figure 8 exhibits the optical response of CA, Mg_3_(VO_4_)_2_@CA, MgO@CA, Mg_3_(VO_4_)_2_/MgO@CA, and Mg_3_(VO_4_)_2_/MgO/GO@CA films. The absorption behavior validates the idea that the oxides insertions lead to peaks with higher intensities in the absorption pattern. Thus, it is clear that Mg_3_(VO_4_)_2_, MgO, and GO show higher positions than the CA spectrum. The peak positioned at 224.5 nm is a pointer to *n*→π* transition. As well as this, the CA polymer spectrum is almost a flat-horizontal line after 285 nm [66]. Regarding the previous studies, the Mg_3_(VO_4_)_2_ has a negligible visible light absorption ability in comparison with other magnesium vanadate species, with a band gap of 3.1 eV as in Table 2 [67,68]. Chayed et al., 2011 proposed a MgO band gap average of 5.8 eV with different methods of preparations. Concerning these facts and tabulated data, the fabricated films cause a significant reduction in band gap in comparison with a pure MgO band gap The absorption coefficients (*α*) are detected via Beer–Lambert’s formula $\alpha \left(\lambda \right)=\frac{2.303A}{d}$, where *A* is absorbance and *d* is film’s thickness. 

As shown in Figure 8, *α* is measured as a function of (photon energy *h*ν) and it is obvious that the absorption edge is moving along the x-axis with varying mixed oxides amount and types. It is 4.9 eV for polymeric matrix (CA), and then diminishes to 3.45 eV for Mg_3_(VO_4_)_2_/MgO/GO@CA. Additionally, the band-gap could be calculated using the next equation $\alpha h\nu ={A(h\nu -Eg)}^{m}$, where *h*ν is the photon energy, while *E_g_* is the band-gap and *A* refers to the band tailing parameter. The power (*m*) refers to the type of transition; where (indirect if *m* = 2, and direct if *m* = 0.5). The diminishing of bandgap reflects raising in crystallographic ordering. On the other hand, the refractive index (*n*) could be found via Dimitrov and Sakka equation as a function of indirect energy bandgap as follows: $\frac{n^{2}-1}{n^{2}+1}=1-\sqrt{\frac{{Eg}_{i}}{20}}$. The refractive index raises from 1.73 for CA to 1.81 for Mg_3_ (VO_4_)_2_/MgO/GO@CA film. Thus, the manipulation of refractive index alteration is dependent on the film’s chemical composition, as proven in Table 2.

### 3.7. Raman Spectroscopy 

Raman spectroscopy is thought to be a particularly flexible optical technique for the analysis of graphitic materials. The Raman spectra of Mg_3_(VO_4_)_2_/MgO/GO@CA are illustrated in Figure 9. GO characteristic peaks are presented in1358 cm^−1^, 1598 cm^−1^, and 2720 cm^−1^ which refers to D and G and 2D bands [69,70]. Meanwhile, the characteristic bands of CA 658 cm^−1^, 906 cm^−1^, 1122 cm^−1^, 1392 cm^−1^, 1440 cm^−1^, 1739 cm^−1^, and 2935 cm^−1^ are referred to as C-OH bond, C-H bond, stretching of C-O-C glycosidic linkage, C-H bond which is in the acetyl groups, asymmetric vibration of C-H, carbonyl group (C=O), and C-H stretching, respectively [71]. On the other hand, the main characteristic peaks of MgO are presented as 1088 and 2884 cm^−1^ [72]. Furthermore, the bands of VO_4_ are found in 820 cm^−1^ and 880 cm^−1^ [73].

### 3.8. Ionic Release

The concentrations of ions that could be released from the film containing Mg_3_ (VO_4_)_2_/MgO/GO@CA were studied. It was found that the concentration of carbon reached around 1430 ± 38 ppm, while the magnesium reached around 46 ± 5 ppm, and the vanadium reached about 35 ± 7 ppm. The nitrogen has not been detected through the composition. The low concentrations of these ions indicate their low cytotoxicity. On the other hand, carbon is common between graphene oxide and cellulose acetate. Thus, the trace magnesium and vanadium are suggested to release through the surrounding environment at a slow rate and low content, which avoids their unwanted effects that might be provoked by the high concentrations.

## 4. Conclusions

Multifunctional nano-films are fabricated by using cellulose acetate/magnesium ortho-vanadate (MOV)/magnesium oxide/graphene oxide for wound healing applications. SEM micrograph of Mg_3_(VO_4_)_2_/MgO/GO@CA film introduces the porous film with flattened rounded MgO grains with an average size of 0.31 µm. Concerning wettability, the binary composition of Mg_3_(VO_4_)_2_@CA occupies the lowest contact angle at 30.15 ± 0.8°, while pure CA represents the uppermost one at 47.35 ± 0.4°. The cell viability ratio with the usage of 4.9 µg/mL of Mg_3_(VO_4_)_2_/MgO/GO@CA is 95.77 ± 3.2%, while a higher concentration of 5000 µg/mL exhibits viability of 19.23 ± 3.1%. Optically, the refractive index flew from 1.73 for pure polymer CA to 1.81 for Mg_3_(VO_4_)_2_/MgO/GO@CA film. The thermal analysis showed three main degradation regions. The initial temperature of the first stage was 30 °C and the final temperature was 289 °C. The peak temperature of this stage was 64.5 °C with a 13% Wt. loss. On the other hand, the second stage existed in the range from 289 to 375 °C with a weight loss of 52%. Furthermore, the last stage occurred from 375 to 472 °C with a Wt. loss of 19%. The residue was about 16% from the initial Wt. The obtained results exhibit improvements in structural, thermal, and optical integrity that offer superior biocompatibility.

## Figures and Tables

**Figure 1 materials-16-03009-f001:**
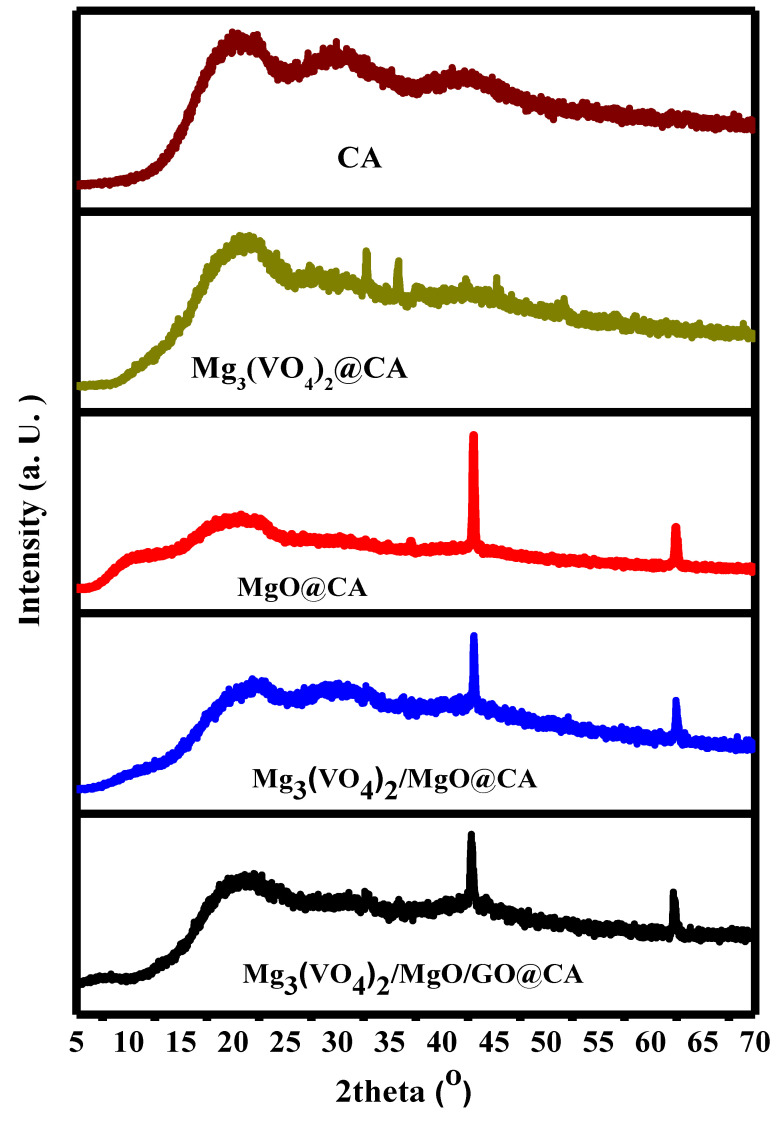
XRD analysis of CA-based films.

**Figure 2 materials-16-03009-f002:**
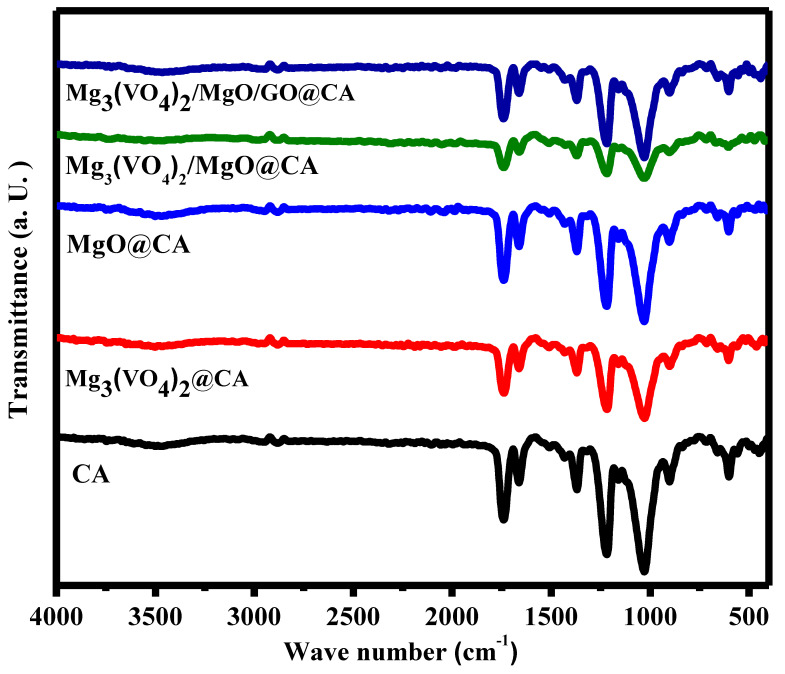
FTIR analysis of CA-based films.

**Figure 3 materials-16-03009-f003:**
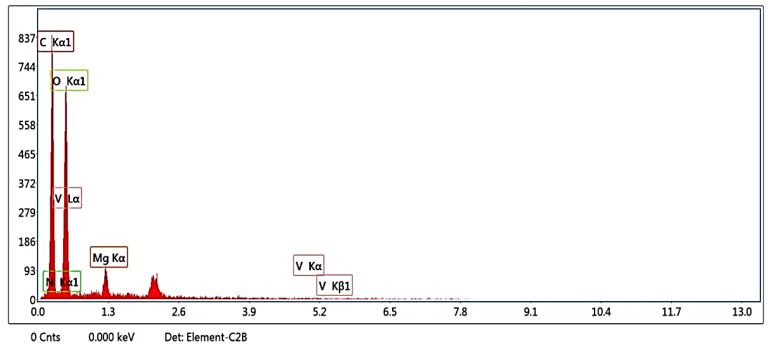
EDAX of Mg_3_(VO_4_)_2_/MgO/GO@CA film.

**Figure 4 materials-16-03009-f004:**
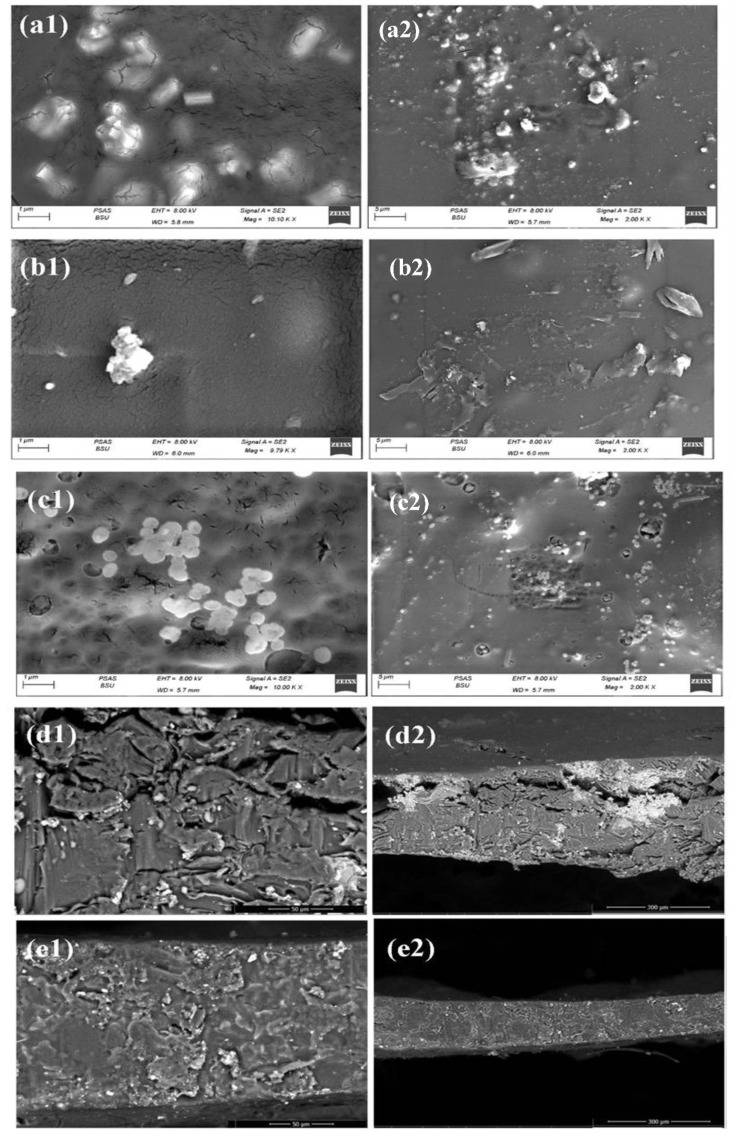
SEM of CA films with a variation of dopants where (**a1**,**a2**) Mg_3_(VO_4_)_2_@CA film, (**b1**,**b2**) Mg_3_ (VO_4_)_2_/MgO@CA, (**c1**,**c2**) Mg_3_ (VO_4_)_2_/MgO/GO@CA, (**d1**,**d2**) the cross-section of Mg_3_ (VO_4_)_2_/MgO@CA, (**e1**,**e2**) the cross-section of Mg_3_ (VO_4_)_2_/MgO/GO@CA.

**Figure 5 materials-16-03009-f005:**
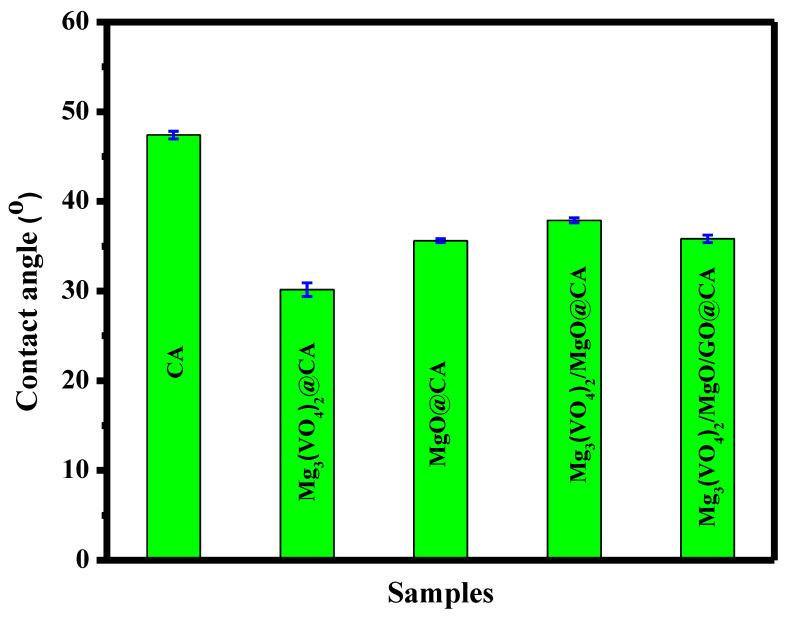
Contact angle histogram for CA and modified films.

**Figure 6 materials-16-03009-f006:**
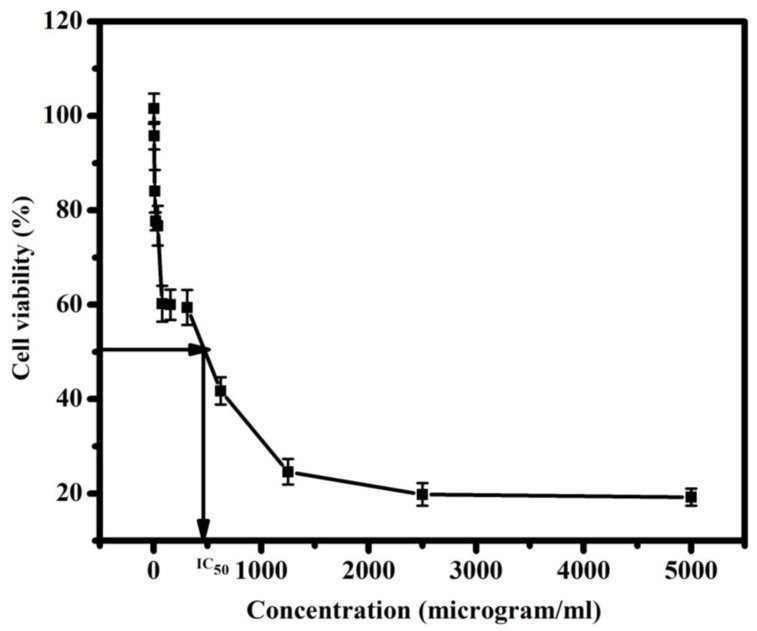
Cell viability of Mg_3_(VO_4_)_2_/MgO/GO@CA, using normal lung cell line.

**Figure 7 materials-16-03009-f007:**
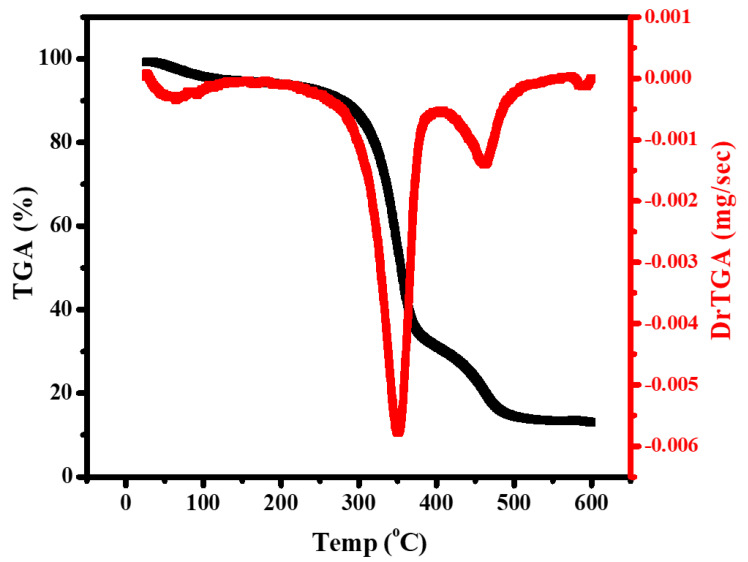
Thermal behavior of Mg_3_(VO_4_)_2_/MgO/GO@CA film.

**Figure 8 materials-16-03009-f008:**
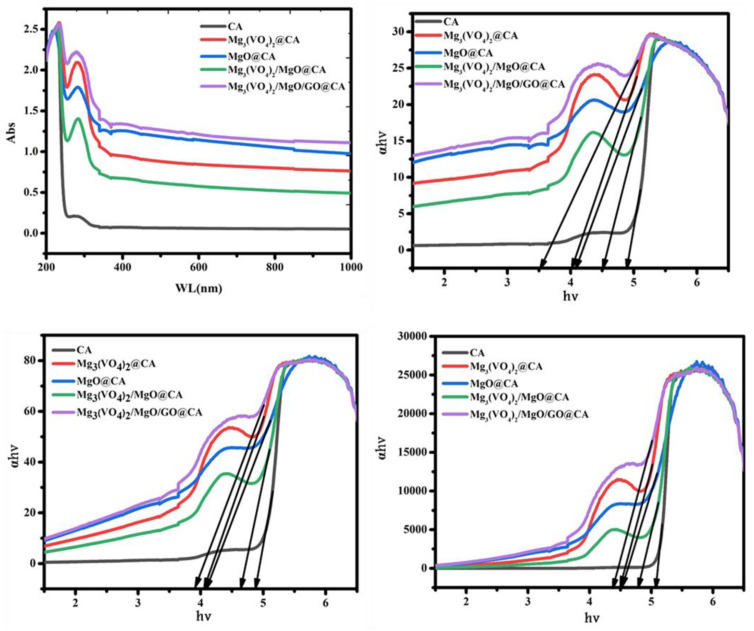
Optical behavior of the CA-based films.

**Figure 9 materials-16-03009-f009:**
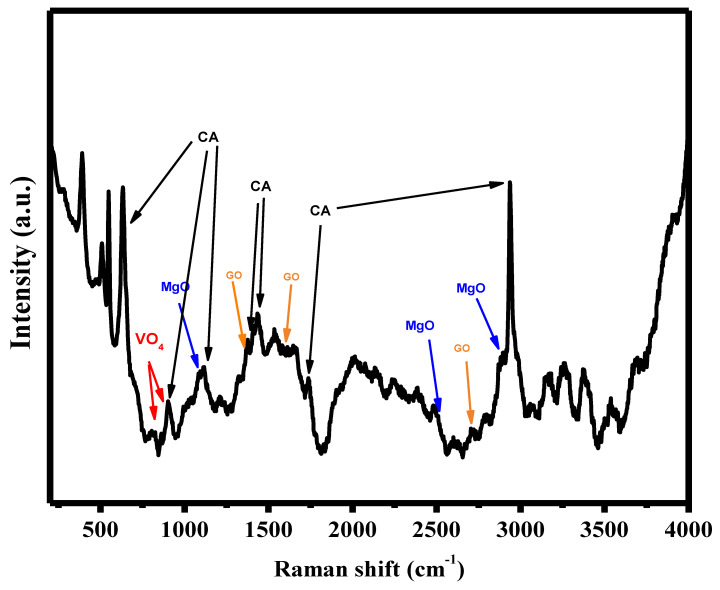
The Raman shift for Mg_3_(VO_4_)_2_/MgO/GO@CA.

**Table 1 materials-16-03009-t001:** EDX qualitative and quantitative data of Mg_3_(VO_4_)_2_/MgO/GO@CA.

Element	Weight %	Atomic %
C K	43.05	50.66
N K	2.56	2.59
O K	50.68	44.77
MgK	3.12	1.81
V K	0.59	0.16

**Table 2 materials-16-03009-t002:** Optical properties of CA-based nano-films, including absorption edge, direct and indirect bandgaps, and refractive index (*n*).

Composition	Thickness (mm)	Absorption Edge (eV)	Bandgap (eV)		*n*
			Direct	Indirect	
CA	0.1	4.9	4.9	5.1	1.73
Mg_3_(VO_4_)_2_@CA	0.3	4.05	4.05	4.5	1.8
MgO@CA	0.25	4.1	4.1	4.55	1.78
Mg_3_(VO_4_)_2_/MgO@CA	0.12	4.5	4.7	4.75	1.75
Mg_3_(VO_4_)_2_/MgO/GO@CA	0.31	3.45	3.9	4.4	1.81

## Data Availability

The raw/processed data generated in this work are available upon request from the corresponding author.

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
