# Peer review of "Magnesium Ortho-Vanadate/Magnesium Oxide/Graphene Oxide Embedded through Cellulose Acetate-Based Films for Wound Healing Applications"

_materials, 2023, doi:10.3390/ma16083009_

Round 1

Reviewer 1 Report

1. The authors should provide the Raman spectroscopy studies to confirm the GO structure.

2. The EDX is only the surface-sensitive technique. To provide the quantitive element analysis, the authors should use the ICP-OES technique (Mg, and V) and elemental analysis CHNO (C, N and O). 

3. The SEM image of the thin film cross-section will be helpful to layer quality.  

Reviewer 2 Report

Taher and group propose the research on the “Magnesium ortho-vanadate (MOV)/ magnesium oxide/ graphene oxide embedded through cellulose acetate-based films for wound dressing”. However, there are some issues with the current research paper that need to be included and/or clarified before the publication, as detailed below:

1.     The author should exclude the short form used in the title of this manuscript.

2.     The manuscript must be checked thoroughly for grammatical and typing mistakes, as there are many mistakes throughout the manuscript.

3.     Why do authors use multiple nanomaterials in this formulation for wound healing? Is it over-exploiting similar kinds of nanomaterials for wound healing?

4.     What are the hypothesis and possible mechanisms of this formulation for expediting the wound healing process?

5.     The authors use a very high concentration of nanomaterials in the formulations. Is it possible that the materials were found to be biocompatible? Please check the data properly.

6.     The author must statistically analyze the contact angle and cell viability.

7.     What are the main novelty and future prospects of this study?

8.     Which types of problems were faced during the fabrication of this formulation?

Author Response

Reviewer 2

Taher and group propose the research on the “Magnesium ortho-vanadate (MOV)/ magnesium oxide/ graphene oxide embedded through cellulose acetate-based films for wound dressing”. However, there are some issues with the current research paper that need to be included and/or clarified before the publication, as detailed below:

  1. The author should exclude the short form used in the title of this manuscript.

Response:

The abbreviation in the title has been removed as follows:

Magnesium ortho-vanadate/magnesium oxide/graphene oxide embedded through cellulose acetate-based films for wound healing applicaation

  1. The manuscript must be checked thoroughly for grammatical and typing mistakes, as there are many mistakes throughout the manuscript.

Response:

The English language has been revised carefully through the whole manuscript.

  1. Why do authors use multiple nanomaterials in this formulation for wound healing? Is it over-exploiting similar kinds of nanomaterials for wound healing?

Response:

An additional statement has been added to clarify that as follows:

Several nanoparticles have been added to the CA to enhance its properties and also to give it more several properties. Mg is chosen to be additive in the membrane because it has appropriate mechanical properties and enhances the catalytic activity of many enzymes. On the other hand, the presence of vanadate in the scaffold is due to its ability to enhance the interaction with cellular proteins and its antibacterial activity. Meanwhile, GO exhibits suitable properties for wound healing applications such as its antibacterial activity, drug-loading capacity, mechanical properties, electrical properties, and hydrophilic behavior.

  1. What are the hypothesis and possible mechanisms of this formulation for expediting the wound healing process?

Response:

An additional paragraph has been added to clarify the mechanism of healing as follows:

Polymeric films play a significant role in the healing process as it is the matrix in which the nanoparticles are encapsulated. When the fabricated film is placed over the wounded area, the film releases the nanoparticles. These nanoparticles play a significant role in cell proliferation and cell growth. On the other hand, bacterial invasion plays a key role in delaying the healing time so the encapsulating nanoparticles play antibacterial activity is important to accelerate the healing process. Further, the addition of nanoparticles and the porosity of the films play a vital role in decreasing the contact angle which means that the hydrophilic nature of the film is increased. This leads to an increase in the chemical activity of the film and increases in cell attachment which accelerates the healing process.

  1. The authors use a very high concentration of nanomaterials in the formulations. Is it possible that the materials were found to be biocompatible? Please check the data properly.

Response:

An additional paragraph has been added to clarify that as follows:

The concentration of the nanoparticles compared to the polymeric film is very low as the total concentration of the nanoparticles is 0.25 g. the ratio between the nanoparticles and the film is 0.25/2 w/w. This low concentration of nanoparticles is not toxic to the human body. To make sure that the sample is not toxic to the human body, the films are cultured to test the cell viability. According to the previous results of cell viability the cells not only safe and alive but also they are growing and proliferating. This means that the concentration of nanoparticles is safe for the human body.

  1. The author must statistically analyze the contact angle and cell viability.

Response:

An additional paragraph has been added to clarify that as follows :

The wettability of the films has been measured using a custom system. About 2 cm x 2 cm from the sample was placed in a holder in front of the camera. Then a water droplet was dropped over the sample the image was taken by (HiView) application. Using this program the contact angle was measured by drawing a baseline measuring the angle from left and right. The average and standard deviation was calculated using excel functions.

 The viability of the cells was assessed using an optical analyzer. The data of cell viability was drawn using the origin lab and the viable cells ratios were taken from the graph.

  1. What are the main novelty and future prospects of this study?

Response:

An additional paragraph has been added as follows:

Several nanoparticles have been added to the CA to enhance its properties and also to give it more several properties. Mg is chosen to be additive in the membrane because it has appropriate mechanical properties and enhances the catalytic activity of many enzymes. On the other hand, the presence of vanadate in the scaffold is due to its ability to enhance the interaction with cellular proteins and its antibacterial activity. Meanwhile, GO exhibits suitable properties for wound healing applications such as its antibacterial activity, drug-loading capacity, mechanical properties, electrical properties, and hydrophilic behavior. These unique properties of these nanoparticles made it a good choice to be embedded in the CA membrane for wound healing applications.

  1. Which types of problems were faced during the fabrication of this formulation?

Response:

An additional paragraph has been added as follows

Although this scaffold has unique properties, the chemical stability of the scaffold could not be controlled. The rate of degradation of the scaffold and drug release rate is not adjustable. It is very important to adjust the ionic release dose as the high dose may cause cytotoxicity while the low released dose may have a weak effect on the healing process. So it is very important to adjust the release the rate of ionic release.

Round 2

Reviewer 1 Report

The manuscript can be aproved in present form.

Author Response

Many thanks for your comments that improving the quality of our manuscript

Reviewer 2 Report

The authors tried to attempt all queries and suggestions. But, statistical analysis and the addition of error bars and significant difference were remains pending in contact angle and cytotoxicity results. The authors must revise the manuscript and perform the statistical analysis properly.

Author Response

The authors tried to attempt all queries and suggestions. But, statistical analysis and the addition 
of error bars and significant difference were remains pending in contact angle and cytotoxicity 
results. The authors must revise the manuscript and perform the statistical analysis properly.
minor revision
Response: The standard deviation has been computed and figure 5 has been re-drawn to contain 
the error bars to show the statistical calculations of the contact angle results.
The cell viability test was already repeated three times. Now, the results have been introduced 
and plotted through Fig. 6 to show the standard deviation values, which have been modified into 
the results and discussion section
